# Development of HPLC Method for Simultaneous Determination of Ibuprofen and Chlorpheniramine Maleate

**Hasan Aldewachi [1,*] and Thamer A. Omar [2,3]**

1   College of Pharmacy, Ninevah University, Mosul 41002, Iraq
2   College of Pharmacy, University of Mosul, Mosul 41002, Iraq
3   Department of Chemical and Biochemical Engineering, Rutgers University, Piscataway, NJ 08854, USA
*   Correspondence: hasan.saad@uoninevah.edu.iq

**Abstract:** One of the most prevalent over-the-counter cold and cough medications is the chlorpheniramine maleate (CPM)–ibuprofen (IBF) combination. A reversed-phase high-performance liquid chromatography (RP-HPLC) method was effectively optimized and developed for the simultaneous detection of chlorpheniramine maleate and ibuprofen in a pharmaceutical formulation. The mobile phase for the RP-HPLC method was an isocratic combination of acetonitrile and 0.01 M acetate buffer at pH 3.8 (55:45; $v/v$) on an Eclipse Plus C18 reversed phase column. An ultraviolet (UV) detector with a wavelength of 225 nm was used to detect the analytes at a flow rate of 1.0 mL/min. CPM and IBF were satisfactorily eluted, with mean retention times of 2.09 and 6.27 min, respectively. The approach was shown to be linear ($R^2 > 0.9998$ for CPM and 0.9992 for IBF), precise (% *RSD* 3.02% for CPM and 3.48% for IBF), accurate (% recoveries 97.7–98.9% for CPM and 101–104.5% for IBF), specific, easy to use, sensitive, quick, and robust. Limits of detection (LODs) were found to be 10 and 27 µg/mL for CPM and IBF, respectively. Without interference from excipients, the validated method could be utilized in regular quality control analysis of various dosage combinations of hard gelatin capsules containing CPM and IBF.

**Keywords:** RP-HPLC; ibuprofen; chlorpheniramine maleate; pharmaceutical formulation

## 1. Introduction

Medicinal combinations are formulations containing one or more active pharmaceutical ingredients (APIs) intended for use in a fixed-dosage form. The majority of multi-component dosage forms have two or more active components, each of which contributes to the overall therapeutic efficacy of the medication. When the selected agents have diverse modes of action that give additive or synergistic effectiveness, this idea is advantageous [1].

However, most official pharmacopoeia monographs are for individual component drugs, so local pharmaceutical manufacturing companies use methods that involve multiple and repeated extractions to separate each active component before quantification using spectrophotometry or titrimetry in the analysis of multi-component drug formulations. As a result, such procedures are time-consuming and inconvenient. Consequently, researchers have developed a variety of analytical methods to facilitate the simple and speedy testing of multi-component dosage forms. With chromatographic techniques being the preferred analytical approach, numerous researchers have labored to establish several liquid chromatographic methods for the simultaneous estimation of multiple active components in multi-component medicines [2–5].

Chlorpheniramine maleate (CPM) and ibuprofen (IBF) are available in various multi-component formulations in the market either as the only active ingredients or as part of other active ingredients that make up the drug products. These medications are typically used to treat sneezing, itching, watery eyes, runny nose, headache, and pain or fever caused by allergies or the common cold [6]. Because of this, researchers have been trying to come

up with methods that can simultaneously identify and quantify some or all of the active components that are present in such formulations. Figure 1 shows the chemical structures of IBF and CPM with their pKa values.

(a)

pKa = 2

(b)

pKa = 9.2

**Figure 1.** Chemical structures of the two analytes (**a**) ibuprofen and (**b**) chlorpheniramine maleate.

IBF is a propionic-acid-based non-steroidal anti-inflammatory drug. (RS)-2-(4-isobutyl phenyl) propionic acid is the chemical name. CPM is an antihistamine used for allergic conditions and is used to treat allergies, hay fever, and common cold symptoms. It is 3-(4-chlorophenyl)-N, N-dimethyl-3-pyridin-2-ylpropan-1-amine. Spectrophotometric, spectrofluorometric, high-performance liquid chromatography (HPLC), gas chromatography (GC), and high-performance thin liquid chromatography (HPTLC) techniques for estimating IBF alone or in combination with other drugs in pharmaceutical formulations and biological fluids have been discussed in the literature [7–12].

CPM has been measured in pharmaceutical formulations and biological fluids by RP- HPLC [13], HPTLC [14,15], capillary electrophoresis [16] and spectrophotometric methods [17,18] separately or in combination with other medicines. For simultaneous measurement of these medicines in tablet formulation, spectrophotometric techniques have been published [18]. In this study, a successful effort was made to estimate both of these medications concurrently using a cost-effective and time-saving HPLC technique.

Several chromatographic methods for the determination of both IBF and CPM in certain formulations, sometimes with other different ingredients, have been developed [19–21]. All these methods require relatively long analysis times ranging from 15 to 30 min, rendering them unsuitable for routine analysis. Therefore, there is a pressing need to develop more convenient methods to provide shorter analysis times with a satisfactory limit of detection and determination.

The majority of reported HPLC methods used C8 or C18 bonded silica columns, a mobile phase of acetonitrile (ACN) and phosphate buffer in various ratios, and UV detection. The chromatographic elution of ionizable substances on such columns is dependent on the mobile phase pH factor, which has been validated for IBF and CPM, whose elution is proportional to their degree of ionization [22]. IBF elutes mostly in its undissociated form at acidic pH (pKa $_{IBF}$ = 2), which explains the increase in retention of this compound in a C8 or C18 type column due to hydrophobic contact. As a result, IBF requires a minimum of 50% ACN to elute in a short period of time [23]. CPM is unprotonated at its amino group

and weakly retained at the same pH levels (pKa CPM = 9.2), even when the ACN ratio is near low (less than 10%) [24]. Being formulated as a salt, CPM might express two peaks for chlorpheniramine (CP) as the active constituent and the other for malic acid [25].

In fact, both IBF and CPM are used as model drugs in this study. Because CPM is a basic drug and IBF is an acidic drug, it was of interest to investigate and develop an HPLC method that can accurately determine the quantity of these two distinct drugs. It was also necessary to investigate how changing the polarity and pH of the mobile phase could affect the separation and detection of these two drugs. As a result, when selecting these two drugs, we concentrated on their physical and chemical properties. To a lesser extent, we considered that the combination of these two drugs could be available in some over-the-counter (OTC) drugs to treat common cold symptoms.

This study aims to establish a fast, simple, precise, and reproducible RP-HPLC method for the quantitative measurement of CPM and IBF. Testing the validity of the developed method with at least one additive microcrystalline cellulose (MCC) and how this developed HPLC method can detect the homogeneity of these two drugs within MCC when making the hard gelatin capsules (HGCs) is another goal for the study.

## 2. Materials and Methods

### 2.1. Materials

IBF pure powder was purchased from Spectrum Chemical Mfg. Corp. (New Brunswick, NJ, USA) while CPM pure powder was obtained from Encore Scientific (Broken Arrow, OK, USA).

VWR Life Science (Solon, OH, USA) was the supplier for both methanol HPLC grade and sodium acetate. Acetonitrile of HPLC grade was obtained from EMD Millipore Corp. (Taunton, MA, USA). Glacial acetic acid was bought from Avantor Performance Materials, LL. (Allentown, PA, USA). The water was distilled and deionized using the Millipore Milli Q Ultrapure technology. MCC(Avicel®PH-200) purchased from DuPont Nutrition USA, Inc. (Wilmington, DE, USA).

### 2.2. Standard Solution Preparation

Accurately weighed 10 mg CPM and 20 mg IBF working standards were transferred into a 100 mL volumetric flask and adjusted with methanol to volume to yield a CPM and IBF stock standard solution. After adding 100 mL of methanol, the prepared solutions were sonicated for 5 min. With the diluent, the solution was brought to its final volume. Then, 2 mL of CPM stock standard solution and 2 mL of IBF stock standard solution were transferred into a 100 mL volumetric flask, and the solution was brought up to final volume with the diluent (C IBF = 0.2 mg/mL, CPM = 0.1 mg/mL).

Acetate buffer (0.01 mol/L) was prepared by dissolving an accurately weighed 577.2 mg of monobasic sodium acetate into 1000 mL of water, then pH was adjusted with acetic acid to the various pHs tested, mixed well, and the solution was filtered and degassed.

### 2.3. Chromatographic Conditions

The Eclipse Plus C 18 (Agilent Zobrax) column (150 mm × 4.6 mm) was equilibrated with mobile phase acetonitrile:acetate buffer (ACN:AcONH$_4$). The mobile phase ratios (MPRs) were changed at different levels to select the best MPRs. The pH of the acetate buffer was changed from 6.8 to 3.8 to explore the effect of the pH on the sharpness of the peak and how the different ionization profiles of the drugs could affect the peak properties and was adjusted with acetate buffer to the pH 3.8. The flow rate was maintained at 1 mL/min, eluents were monitored with an ultraviolet (UV) detector at different wavelengths (215, 225, and 260) nm, and the injection volume was 20 μL. The total run time was 10 min.

### 2.4. Preparation of Physical Blend and HGCs

A fifty-gram blend of 10% CPM, 20% IBF, and 70% MCC (Avicel pH 102) was prepared using a laboratory scale Resonant Acoustic® Mixer (RAM, Resodyn Acoustic Mixers, Butte,

MT, USA), which utilizes low frequency and high-intensity acoustic energy to enhance mixing for small-scale blends. The blend was prepared at 40% intensity for 2 min of vibration time. Then, 400 mg of these blends was manually filled into all of the 40 HGCs (size 0).

### 2.5. Sample Preparation

Eight HGCs were randomly selected. Each capsule was opened and its contents were transferred into a 100 mL volumetric flask. Methanol was added to the 100 mL mark. Then, these samples were sonicated for 50 min and left overnight to ensure complete extraction of CPM and IBF. Then, HPLC was conducted to measure the amount of CPM and IBF in these samples.

### 2.6. Validation of the Method Parameters

The method was validated by following the analytical performance parameters recommended by the International Conference on Harmonization (ICH) [26]. Calibration curves were constructed using standard solutions of CPM at 0.2, 0.3, 0.4, 0.5, and 0.6 mg/mL while standard solutions of IBF were made at 0.4, 0.6, 0.8, 1, and 1.2 mg/mL. The percentage of CPM, % of IBF, and relative standard deviation (*RSD*%) were calculated for these samples. The LOD and LOQ were based on the standard deviation of the response and the slope of the corresponding curve using the following equations: LOD = 3 σ/S; LOQ = 10 σ/S where σ is the standard deviation (SD) of the intercept and S is the slope of the related calibration graphs. The *RSD* of the analytical method estimates the ratio between the standard deviation and the mean of a sampled population [27]. *RSD* can be calculated as:

$$RSD\% = \frac{Standard\ Deviation}{Mean} \times 100$$

This value is very important to measure the drug variability in blend samples (blend uniformity).

## 3. Results and Discussion

The reversed-phase LC method discussed in this work was created to offer a quick way to check the CPM and IBU levels in capsules. It employs a simple mobile phase. Using the prescribed chromatographic conditions, all samples were examined. The test method's range was established at 75–125% of the final product's label claim.

### 3.1. Optimization of the Mobile Phase

Changes in the mobile phase's pH and composition have an impact on the chromatographic parameters tailing factor, capacity factor, resolution, and separation efficiency. The goal of optimizing the mobile phase was to reduce the asymmetry factor in chromatographic peaks for active medicinal components. There were several MPRs that attempted to resolve all three chromatographic peaks, the first of which used simple AcN: AcONH$_4$ (40:60, *v/v*), AcN: AcONH$_4$ (45:55, *v/v*), AcN:AcONH$_4$ (50:50, *v/v*) and AcN:AcONH$_4$ (55:45, *v/v*). Satisfactory chromatographic separation and broadness of peaks were obtained with an optimal ratio of 45:55 AcN: AcONH$_4$. So, AcN: (0.01 M) AcONH$_4$ (55:45, *v/v*) (pH: 6.8) resolved the merged IBF and dissociated chlorpheniramine and malic acid peaks with retention times of less than 7 min.

Compared to previous LC methods, this method provides the shortest analysis time for a combination of IBF and CPM and this may be due to the higher affinity of the analytes for the ACN:AcONH$_4$ buffer. Table 1 provides insight and comparison between our developed methods and other chromatographic methods. Overlain UV spectra of the drugs showed that CPM and IBF absorbed appreciably at 225 nm, so detection was carried out at 225 nm (Figure 2). It can be seen that almost all of the methods share a comparable stationary phase (C18 column) but differ in their mobile phase composition.

**Table 1.** Comparison of chromatographic methods in terms of analysis time, HPLC run parameters.

| Method (Year) | Run Time | Mobile Phase | Column & Gradient |
|---|---|---|---|
| Sanchaniya et al. [19] | 14 min | Acetonitrile : methanol : phoshphate buffer (50:20:30, *v/v/v*) | C 18 column (5 μm × 250 mm × 4.6 mm) isocratic |
| Asçi et al. [20] | 15 min | Acetonitrile buffer (15:85) for 5.5 min, (45:55) for 5.5–12 min, (60:40) for 12–17 min. | C18 (300 mm × 3.9 mm) gradient |
| Sarılmışer et al. [21] | 28 min | Buffer: methanol (80:20) | ODS4 column (250 mm × 4.6 mm; 5 μm) isocratic |
| This method | 7 min | Acetonitrile and acetate buffer at pH 3.8 (55:45; *v/v*) | Eclipse Plus C18 isocratic |

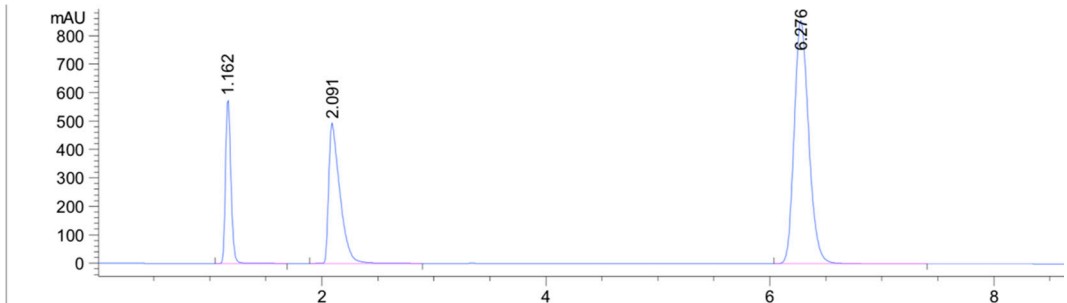

**Figure 2.** UV-detected chromatogram of the capsule at a wavelength of 225 nm. Malic acid detected at 1.16 min, chlorpheniramine (1 μg/mL) at 2.09 min, and ibuprofen (1 μg/mL) at 6.27 min.

*3.2. Method Validation*

The developed method was also validated according to validation parameters comprised of accuracy (% of recovery), precision (% *RSD*), linearity (regression coefficient), LOD, and LOQ. Table 2 summarizes the results of validation parameters. Furthermore, the resolution, tailing factor, and theoretical plates complied with standard requirements (ICH guidelines).

**Table 2.** Method validation results for CPM and IBF.

| Parameters | CPM | IBF |
|---|---|---|
| Linearity (range) (mg/mL) | 0.3–3.5 | 0.4–4.0 |
| Retention time (min) | 2.09 | 6.27 |
| Detection limit (μg/mL) | 10 | 27 |
| Quantitation limit (μg/mL) | 33 | 90 |
| Accuracy (%) (*n* = 5) | 97.7–98.9 | 101.0–104.5 |
| Intra day precision. (*n* = 5) | 1.50 | 1.35 |
| Day to day precision (*RSD*%) * (*n* = 5) | 2.14 | 3.48 |

* *RSD* is relative standard deviation and *n* is number of determinations. CPM—chlorpheniramine maleate; IBF—ibuprofen.

The approach's accuracy was determined by using the conventional addition method to calculate CPM and IBF recoveries. CPM and IBF recovery rates were determined to be 97.7–98.9% and 101–104.5%, respectively. Using five replicates of three distinct concentration levels (0.5, 1, and 2 μg/mL), the method's precision was determined for intraday (one-day procedure under stable conditions) and interday (three separate days) changes. Obtained accuracy and precision results were comparable to accuracy and recovery reported in previous literature reports [19–21]. Table 2 summarizes the relative standard deviation and recovery of each component.

The values of LOD and LOQ are also given in Table 3. The LODs and LOQs were determined as 27 and 90; for IBF 10; and 33 μg/mL for CPM, respectively. Although lower LODs and LOQs were achieved in previous reports, these values are extremely low and

consequently the procedure is highly sensitive and can be used to detect both APIs within their normal levels in pharmaceutical dosage forms.

**Table 3.** Linearity data for calibration graphs for the simultaneous determination of IBF and CPM by developed HPLC method.

| Drug | Concentration Range, mg/mL | *Correlation Coefficient* |
|---|---|---|
| *Ibuprofen* | *0.4–4.0* | *0.9998* |
| *Chlorpheniramine* | *0.3–3.5* | *0.9992* |
| *Malic acid* | *0.3–3.2* | *0.9970* |

The calibration curve for CPM was found to be linear with a correlation coefficient of 0.9992 in the range of 0.3–3.5 mg/mL. In the range of 0.4–4.0 mg/mL, the calibration curve for IBF was found to be linear, with a correlation value of 0.9998. With a correlation value of 0.9970, the calibration curve for maleate was found to be linear in the range of 0.3–3.2 mg/mL (Table 3). Linearity and $R^2$ values achieved by this method were comparable and sometimes superior to linearity results obtained by former reports for determination of CPM and IBF.

Injection repeatability tests were used to assess instrument precision (%). The method's precision is indicated by the low *RSD* readings. Within a day, each sample received three replicate injections, and the mean concentrations were calculated. Using the same considerations of the APIs as in the intra-day precision, the inter-day accuracy was assessed over the course of three consecutive days ($n = 5$). Relative standard deviations (*RSD*) were estimated based on the concentrations of the two APIs, and the *RSD* values for CPM and IBF were found to be 3.02 to 3.48%. The limit established for the precision examination of the instrumental system demonstrated that the created methods developed well-functioning equipment and that the experimental findings are highly reproducible.

*3.3. Chromatographic Conditions*

As an appropriate approach for quantitative measurement of IBF and CPM, a RP-HPLC method was developed. The chromatographic conditions were modified to produce an effective and straightforward routine procedure. The separation is problematic because the IBF and CPM molecules exhibit opposing basic acidic characteristics.

A preliminary development test was carried out using a C18 column (150 mm × 4.6 mm, i.d., 5 μm) with an acetate buffer/CAN mobile phase. Under the pH conditions studied (between 3.8 and 6.8), the CPM, in its ionized state, invariably emerges out of the dead time, reducing the amount of CAN. For chromatography, isocratic elution with a mobile phase comprising buffer solutions at various pH values and CAN was utilized. CAN did not affect CPM retention time, in contrast to previous reports that showed that concentrations of more than 20% CAN caused CPM to elute during the dead time [27].

The impact of pH on the ionization of IBF and CPM was examined. A 10 mM acetate buffer with a pH of 6.8 was tried first. In respect to the pKa values of both chemicals, this pH value was a compromise. IBF and CPM were both ionized in these circumstances and were not separated. The optimization was continued by adding an acetate buffer to lower the pH.

The HPLC chromatogram showed three distinct peaks owing to the splitting of CPM into chlorpheniramine and maleic acid. Previous literature reports have described peak splitting by heat, UV radiation, pH conditions, and oxidative stress [28,29]. The absorption coefficients at each compound's maximum absorbance (CPM 261 nm = 5760 L/cm mol; IBF 220 nm = 8200 L/cm mol) are comparable, which explains the difference in signal between the two compounds, which is connected to the number of active ingredients in the capsule (200 mg IBF and 2 mg CPM). This distinction is seen in Figure 2.

The proposed reverse phase HPLC technique offers several benefits in terms of mobile phase simplicity, isocratic mode of elution, quick run time, excellent resolution, chemicals availability, and ease of preparation of sample and standard solutions.

### 3.4. Assay of Commercial Formulation (HGC Samples)

CPM and IBF in their combination dose form were effectively determined using the suggested approach. For CPM and IBF (Table 4), the % recovery was determined to range from 96.37 to 102.6 and *RSD* ranged from 2.14 to 3.48%.

**Table 4.** Recovery and relative standard deviation for pharmaceutical capsules.

| Drug | Theoretical % | Measured % | Recovery % |
|---|---|---|---|
| Chlorpheniramine maleate | 10 | 10.85 | 108.50% |
| Ibuprofen | 20 | 18.95 | 94.75% |

The method was applied to the determination of IBF and CPM in a new HGC sample to confirm its practicability and feasibility in the analysis of these components in pharmaceutical preparations. The results of the analysis of the HGC are given in Table 4.

Using peak areas and regression equations derived from the calibration curves, the content (mg) and percentages of each API in the tablet sample were calculated. The manufactured products' mean levels of IBU and CPM fell within the permissible range of 90 to 110 percent of the label quantity according to BP 2020 [30].

### 4. Conclusions

A reversed-phase HPLC approach with UV detection that is accurate, easy, linear, specific, and precise was developed and validated for the simultaneous quantification of chlorpheniramine maleate and ibuprofen.

Various conditions and parameters were studied for the separation of IBF and CPM, at various MPR, detector wavelength, and pH of buffer under isocratic conditions. The verification of the developed HPLC method was done by validation parameters. The results of validation indicated good precision, accuracy, linearity and reliability.

The results revealed that the resolution can be maximized when using pH = 3.8; 1.0 mL/min flow rate and wavelength detection at 225 nm. Thus, the proposed method is rapid, easy to use, and can be used for mundane and quality control analysis. The concentration of CPM and IBF in pharmaceutical dosage form could be satisfactorily determined using the developed isocratic RP-HPLC methods in a short period (less than 7 min).

The proposed method is quick and easy to use for quality control analysis of solid dosage forms that contain CPM and IBF. Using this procedure, HGC containing these two APIs along with one excipient were examined. According to the findings, the total amount of both APIs was within the allowed range and the *RSD* percentage was quite low; this indicated that the product's quality was satisfactory.

**Author Contributions:** Methodology, T.A.O., writing—review and editing, H.A. and T.A.O.; data curation, H.A. and T.A.O.; formal analysis, T.A.O.; investigation, H.A. and T.A.O.; validation, H.A. and T.A.O.; writing—original draft, H.A. and T.A.O.; project administration, T.A.O.; supervision, T.A.O. All authors have read and agreed to the published version of the manuscript.

**Funding:** The research leading to these results received no external funding.

**Institutional Review Board Statement:** Not applicable.

**Informed Consent Statement:** Not applicable.

**Data Availability Statement:** All data are contained within the article.

**Conflicts of Interest:** The authors declare no conflict of interest.

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
