# Peer review of "Development of HPLC Method for Simultaneous Determination of Ibuprofen and Chlorpheniramine Maleate"

_scipharm, doi:10.3390/scipharm90030053_

Round 1
Reviewer 1 Report
The manuscript is not suitable for publications.
There are many typing errors (e.g. upper case in compound name).
English language requires editing throughout manuscript.
Authors used some acronyms without their previous explanation, e.g. RP-HPLC, GC, HPTLC, etc.
Proper discussion is missing.
The main concern is degradation of CPM.
Specific comments and suggestions are given below.
Line 47: There are no pKa values in Figure 1.
Line 56: Please explain ‘’non- aqueous’’ in ‘’ …..using non-aqueous by liquid chromatography….’’.
Lines 56-62: Authors should make clear distinction between separation (e.g. HPLC) and detection (e.g. spectrometry).
Line 85: CPM is missing.
Line 91: Please describe the difference between powder and pure powder.
Line 102: You cannot begin the sentence with a number. Same for the Line 132.
Line 112: Please add manufacturer of the column.
Line 121: How did you choose the proportions in the blend? Please explain or add the reference.
Line 128: How many capsules did you fill?
Line 138: RSD of what? This is unclear.
Line 154-157 and Table 1: For Discussion.
Figure 2: Please add wavelength and concentration determined.
Line 171: Table 1 does not summarize results of validation parameters.
Line 176: Table 1 does not summarize the relative standard deviation and recovery of each component.
Line 172-176: Please add number of replicates and concentration levels for precision and recovery.
Line 179: The values of LOD and LOQ are not given in Table 1.
Line 185: It is unclear how did you measure CPM since maleate was degraded from chlorpheniramine. What exact analytical standards did you use? Degradation should be discussed.
In validation part, it is unclear what is referred to the instrument and what is referred to the method.
Table 2: Suddenly authors mentioned maleate and chlorpheniramine separately. How did you validate it?
Reviewer 2 Report
The manuscript reports the development and validation of the technique for HPLC common analysis of two drugs including in drug formulation. In general, the manuscript demonstrates good analytical results in the field of pharmaceutical analysis and paper can be considered for publication in Scientia Pharmaceutica. Nevertheless, some minor revision and polishing of the English language is required.
Comments:
11. The introduction needs to be updated with fresh literary sources. Unfortunately, in the version presented, only 3 articles were published after 2015, and 23 were published before 2015.
2. Section 2.3.
The column dimensions are listed as “…(3.5 µm ´ 150 mm ´ 4.6 mm)…”. I suppose that column dimensions are (150 mm ´ 4.6 mm) but 3.5 µm is the size of beads. Please, correct the information presented.
3. Page 5, line 178
Please give the equations for LOD and LOQ in Materials and Methods (section 2) in the same way as the equation for RSD. Also, add the reference from which the LOD and RSD equations were taken.
4. Page 6, line 206
The authors provide the dimensions of the column (150 mm ´ 4.6 mm, i.d.) and then provide the value 5 mm. What is it? Size of the beads? Does dimension in mm is correct? Or it must be 5 µm.
5. Table 2.
All values are presented to the first decimal place except two. It looks bad in an analytical article. Please make a uniform representation (0.4-4.0, 0.9970).
6. Table 3.
The same comment as for Table 2. For example, in the first line the values should be presented as: 0.3-3.5 and 0.4-4.0. Or in the fifth line as: 97.7-98.9, 101.0-104.5.
Reviewer 3 Report
The manuscript is interesting but the authors should correct a number of reported errors highlighted in the attached file. In particular, authors should write the structural formulas of the compounds analyzed so that they appear clear and in the same format. Finally, the bibliography should be reported according to the specifications required by the journal

Author Response
We would like to express our gratitude for reviewers for their valuable comments that enriched our manuscript. Here are our response point by point to these questions
Reviewer 3 comments
- the authors should correct a number of reported errors highlighted in the attached file
Thanks for your comments. All reported errors corrected except for capitalization of “maleate” because we removed capitalization from all the compounds according to reviewer 1 suggestion
- authors should write the structural formulas of the compounds analyzed so that they appear clear and in the same format.
Thanks. Structural formula of the analyzed compounds is provided in figure 1
- the bibliography should be reported according to the specifications required by the journal.
Many thanks for alerting us. Required corrections and amendments are made according to journal specifications.

Round 2
Reviewer 1 Report
Discussion section is missing. Is Section 3 actually Results and Discussion?
In general there is a lack of discussion of the results. The authors have limited themselves to presenting the results. Please compare the results of validation with other authors.
Author Response
We would like reviewer 1 for its valuable comments. we realized that our results section really needs to have a proper discussion. We added many paragraphs throughout the results section to discuss validation parameter results and marked these additions highlighted with blue for easy tracking.
Round 3
Reviewer 1 Report
Please see attachment.

Author Response
First of all, we would like to express our gratitude for reviewer 1 valuable comments. We have addressed all of the suggestions apart from exchanging the colon ":" with underscore "_" because we feel that this punctuation mark is suitable and acceptable sign to express ratios.
This manuscript is a resubmission of an earlier submission. The following is a list of the peer review reports and author responses from that submission.
Round 1
Reviewer 1 Report
“Development of HPLC method for simultaneous determination of Ibuprofen and Chlorpheniramaine Maleate” written by Hasan et al. established to determine Ibuprofen and CPM to shorter elution time. Their method able to determine the two analytes shorter time compared to two already published method e.g. https://downloads.hindawi.com/archive/2013/424865.pdf, and https://link.springer.com/article/10.1134/S1061934810070142. As a result, authors study is a well stablished method which goes with the scope of this journal to published. But there are many things’ authors ignore which need to be updated. Therefore I request to editor accept this manuscript after major corrections. Where my corrections are listed below.
1. Please cite line 72, 74, and 79 to establish the claim.
2. Quality of figure 2 and 3 is very poor, need to improve the quality to be a professional figure.
3. Authors should show the chromatogram of sample blank and solvent blank to show the interference of solvent and sample.
4. I like to suggest the authors to add a table and compare their findings with previously published articles.
5. Authors just said sharpness of peak of increased at pH 3.8 than 6.8 but they did not show any chromatograms for comparison. I would suggest adding a chromatogram to establish the claim.
6. Authors just showed percent recovery but did not mention much information on that. Have they used any QC samples (like low, medium, and high)? This information is needed to update to upgrade the manuscript quality.
Reviewer 2 Report
This manuscript has terrible writing and some serious flaws. This reads like an abstract for a conference. At this time, it cannot be accepted for publishing. A chromatographic method development research should often include essential experiments, such as various flow rates, injection volumes, solvent gradients, etc. The study seems to lack methodical execution, in my opinion. Additionally, no strong justification for this work's uniqueness was offered. Also, please address the following specific comments on a resubmission.
1. Abstract – what do you mean by satisfactorily eluted?
2. The spaces between new paragraphs should be eliminated
3. What is the basis for selecting mobile phase chemicals or their mixtures? The solvent systems were arbitrarily selected, and no logical explanation was given
4. The figures look terrible (blurred and poorly presented), and the chromatogram needs to be drawn using graphing software
Reviewer 3 Report
The novelty statement of this work is addressed to the Development of HPLC method for simultaneous determination of Ibuprofen and Chlorpheniramaine Maleate. Unfortunately, the aim of this report is rather than safe. The use of reversed phase high-performance liquid chromatography technique for simultaneous determination of Ibuprofen and Chlorpheniramaine Maleate is well known in several published reports, you can find some published reports as follows:
1- http://dx.doi.org/10.1155/2013/424865
3- Nalini et al., IJPSR, 2014; Vol. 5(2): 410-416.
4- DOI 10.1186/s40064-016-2241-2
5- http://doi.org/10.25135/jcm.9.17.05.039
Moreover, the work does not show any enhancement in the limit of detection as well as the limit of quantification. Consequently, due to the lack of novelty, the paper should be rejected.